# Australian Community Pharmacy Harm-Minimisation Services: Scope for Service Expansion to Improve Healthcare Access

**DOI:** 10.3390/pharmacy9020095

**Published:** 2021-04-26

**Authors:** Sara S. McMillan, Hidy Chan, Laetitia H. Hattingh

**Affiliations:** 1Gold Coast Campus, School of Pharmacy and Pharmacology, Griffith University, Southport 4215, Australia; s.mcmillan@griffith.edu.au; 2The Pharmacy Guild of Australia, Queensland Branch, Brisbane 4000, Australia; hidychy@gmail.com; 3Gold Coast Hospital and Health Service, Southport 4215, Australia

**Keywords:** community pharmacy, harm reduction, interviews, pharmacy practice

## Abstract

Community pharmacies are well positioned to participate in harm-minimisation services to reduce harms caused by both licit and illicit substances. Considering developments in pharmacist practices and the introduction of new professional pharmacy services, we identified a need to explore the contemporary role of community pharmacy in harm minimisation. Semi-structured interviews were undertaken to explore the opinions of stakeholders, pharmacy staff, and clients about the role of community pharmacy in harm minimisation, including provision of current services, experiences, and expectations. Participants (*n* = 28) included 5 stakeholders, 9 consumers, and 14 staff members from seven community pharmacies. Three over-arching themes were identified across the three participants groups: (i) scope and provision, (ii) complexity, and (iii) importance of person-centred advice and support in relation to community pharmacy harm minimisation services. Community pharmacies are valuable healthcare destinations for delivery of harm minimisation services, with scope for service expansion. Further education, support, and remuneration are needed, as well as linkage to other sector providers, in order to ensure that pharmacists and pharmacy staff are well equipped to provide a range of harm minimisation services.

## 1. Introduction

Inappropriate opioid use and opioid-related deaths are of international concern [1,2]; according to the United States of America (USA) Centres for Disease Control and Prevention, increased opioid use has led to the “worst drug overdose epidemic in [their] history” [3]. In Australia, an estimated 150 consumers are hospitalised daily due to opioid abuse; the opioid-related death rate reached 62% of all drug-induced deaths in 2016 [4]. Opioid-related deaths are often associated with pharmaceutical rather than illicit opioids [4]. In 2016–2017, more than 3 million Australians were prescribed opioids, which equated to 15 million subsidised medicines [4]. A United Kingdom (UK) retrospective cohort study identified that just under 15% of consumers newly prescribed opioid treatment for non-cancer pain become long-term users in the first year [5]. Trends of opioid overdoses are variable and fluctuate [6,7]. Although the consequences linked to the abuse of illegal drugs are well-known, the misuse of medicines can also be substantial and is related to short- (e.g., gastrointestinal) or long-term (e.g., psychiatric co-morbidity, dependence) effects, hospitalisation, and mortality [8].

Harm minimisation is defined by the World Health Organisation (WHO) as “a set of policies, programmes, services and actions that aim to reduce the harm to individuals, communities and society related to drugs…” [9]. Harm minimisation services aim to reduce health, social, and economic harms in the community [10]. As medication suppliers, community pharmacies are well positioned to promote and participate in harm minimisation services, particularly considering staff skills and attributes, the widespread set-up and resources of pharmacies, and the extended operating hours of many pharmacies [11,12,13].

A 2012 literature review identified that pharmacists were mainly involved in harm minimisation by offering needle and syringe (NSP) and opioid treatment (OTP) programs [13]. The NSP has been available in Australia since the 1980s [14], and involves provision of sterile needles and syringes and safe disposal methods to reduce the transmission of blood-borne viruses and other infections caused by non-sterile practices [15,16,17]. While low-quality of studies made it difficult to identify the risk reduction of hepatitis C from NSP use [18], an estimated decline in hepatitis C cases of between 15 and 43% during 2000–2010 in Australia has been reported [19]. The OTP has been in place in Australia since the 1970s [20] and aims to reduce opioid misuse through the supply of subsidised opioid medicines, i.e., methadone and buprenorphine, to reduce cravings, relapse risk, and withdrawal symptoms. Generally, the use of opioid agonists in opioid dependence is supported, but this is based on low-to-moderate-quality evidence [21]. The opioid dose is either administered at a pharmacy under pharmacist supervision or may be authorised as take-away doses for consumers stabilised on the treatment [22]. Over 50,000 Australians received such treatment in 2018 [23].

Although the OTP is a crucial harm minimisation strategy, medication dispensing fees are not subsidised by the Australian government and need to be paid by consumers. A Western Australian study highlighted that financial strain could result in the skipping of doses or treatment cessation [24]. Another Australian study conducted in New South Wales and Victoria found that only one-third of pharmacists surveyed (*n* = 669) had consumers that were up-to-date with program payments; inability to pay was a reason for pharmacists to terminate treatment [25]. Other identified barriers include the associated dispensing and record keeping processes that adds to pharmacies’ workload [26,27] and concerns around staff safety [25,28]. A variety of issues have been identified as barriers for NSP users and pharmacists in relation to service implementation. Discrimination and stigma are key obstacles for service users to obtain sterile injecting equipment from community pharmacies [29,30]. Pharmacist religious beliefs [31], fears of attracting drug users, and ethical concerns about supplying needles to these consumers were identified barriers [32].

A more recent community pharmacy harm minimisation role is the over-the-counter (OTC) supply of naloxone, a short-acting opioid antagonist that reverses the effects of opioids to temporarily counteract an overdose [33]. Naloxone was rescheduled in Australia in 2015 to become available as a pharmacy OTC medicine that does not require a prescription [34]. The importance of providing take-home access to naloxone has been advocated for and recognised internationally [35], including in Canada [36], the UK [37], and the USA [38]. However, government subsidy for naloxone through the Australian Pharmaceutical Benefits Scheme (PBS) is only on prescription, which could place a significant cost burden on consumers when obtaining naloxone as an OTC medicine. Several Australian and U.S. studies have identified issues related to naloxone supply, from the perspectives of pharmacy staff [39,40], consumers [41,42], and service providers [43]. This included time restrictions to counsel consumers [40], concerns regarding whether consumers would pay for naloxone in emergency situations [39], and perceived stigma [42]. Staged supply is another harm minimisation service provided by Australian community pharmacies that was introduced in 2011 [44] and funded by the federal government. This service targets consumers with a risk of drug dependency or who have difficulty managing their medication safely [45] to supply medicines in instalments. Little research has been conducted to explore the barriers and facilitators for providing the staged supply service in community pharmacy. There are other public health-related services that pharmacies can provide in relation to minimising harm, e.g., alcohol screening, which was considered as a positive service delivered by community pharmacy [46,47]. However, these services are not remunerated by the Australian government or state departments.

Overall, pharmacists internationally have positive attitudes toward harm minimisation service provision with interest in an increased role, although fear is a key issue: fear of attracting customers who use drugs, losing other customers, and harm to their staff [48,49,50]. Lack of time, knowledge and training, and remuneration, as well as inadequate communication between health providers are other reported barriers [13]. When developments in pharmacists’ practices and the introduction of new pharmacy services were considered, a need was identified to explore the contemporary role of Australian community pharmacy in harm minimisation services. This study aimed to explore the opinions of stakeholders, pharmacy staff, and consumers about the role of community pharmacy in harm minimisation, including the provision of current services, experiences, and expectations.

## 2. Materials and Methods

Qualitative semi-structured interviews with key stakeholders, community pharmacy staff (pharmacists and pharmacy assistants), and current users of harm minimisation services (clients) were conducted between June and November 2018 in southeast Queensland (QLD). A thematic, content analysis approach was used for data analysis. University ethics approval was obtained (GU ref no: 2018/330); the research team referred to the consolidated criteria for reporting qualitative research (COREQ) [51] during study design and implementation.

### 2.1. Study Population

A list of potential interview participants was identified from prior researcher knowledge of the sector, as well as recommendations from study participants, i.e., snowball sampling, a strategy employed in difficult to access groups [52]. For example, community pharmacy staff known to provide NSP or OTP were emailed directly, and key stakeholders approached from various non-government and government organisations that had an active role in harm minimisation services and policy. Study information was provided with consent obtained prior to phone interviews. To increase interview time availability, three researchers (S.M., L.H., and H.C.), all registered pharmacists with experience in conducting semi-structured interviews, were involved in data collection.

Consumer participants were purposively recruited [52] from a not-for-profit health service supporting people who use illicit drugs. Consumer eligibility included being an adult who could provide informed written consent and had experience, or were future users, of a harm minimisation service from a community pharmacy. The harm reduction coordinator from the not-for-profit health service assisted researchers by providing feedback on the information sheet and interview guide, as well as promoting the study to consumers. Consumer interviews were undertaken on-site by one researcher (L.H.) with experience in the drug and alcohol sector; two phone interviews were conducted due to participant preference. Consumer participants were aware that the researcher was a pharmacist and were advised that their decision to participate would not influence their relationship with the health service or their community pharmacy.

### 2.2. Data Collection

To ensure consistency across interviewers and to provide participants with example questions, we developed a semi-structured interview guide (Table 1) tailored for each participant type (stakeholder, pharmacy staff, and client). The interview guide was informed by a literature review [53,54] and tested for face and content validity by an academic pharmacist, a senior clinical mental health pharmacist, and a pharmacist known to the research team. Minor amendments were made, such as wording changes for pharmacy assistants to increase understanding and the addition of further prompts. Key topic areas included the concept of harm minimisation and related services; the role of community pharmacy staff in harm minimisation, including risks, benefits, and barriers; and pharmacy experiences, such as client interactions.

All interviews were audio-recorded with consent, transcribed verbatim, and de-identified using a unique code. Participants were offered an AUD 50 gift voucher in appreciation of their time. Interviews averaged 34.72 min for stakeholders (range: 16.51–46.07), 36.28 min for pharmacy staff (range: 26.20–62.23), and 17.89 min for consumers (range: 13.33–22.39). An external transcribing company transcribed all except two interviews, which were quality checked by a researcher. Participants were offered a copy of their individual transcript for review; two stakeholders and one pharmacist requested this, with no changes made. A written debrief was provided to the entire research team for most interviews to assist with data immersion.

### 2.3. Data Analysis

Two researchers (S.M. and L.H.) were involved in reading and re-reading the transcripts for thematic analysis. Whilst the interview guide was referred to during the data analysis process, an inductive approach allowed for identifying new emerging themes. The interviews were initially coded according to the interview guide by one researcher using the software package NVivo (version 12, QSR International Pty Ltd., Doncaster, Victoria, Australia). The NVivo file was then shared with the other researcher to undertake independent thematic analysis across the entire dataset. This involved the identification of additional themes, which were then discussed in-depth by the two researchers for all three participant groups. These discussions led to the clarification of identified themes and agreement with the higher order themes arising from the coded data. The themes were then compared across participant sub-sets for data triangulation. To ensure credibility and trustworthiness of the data, we provided the final coding framework to the third researcher (HC) who had also read the transcripts for final verification. Because the primary aim of the interviews was to inform survey development, data saturation was neither sought nor achieved.

## 3. Results

Participants included 5 stakeholders, 9 consumers, and 14 pharmacy staff from seven community pharmacies, four of which were independently owned. Most pharmacy staff interviewed were registered pharmacists (*n* = 11), of which six were female and three were pharmacy owners. Stakeholders represented staff and clinicians in policy positions from both government and non-government organisations. Consumer demographic details were not obtained for anonymity purposes.

The following section details the three over-arching themes identified across all participant groups: (i) scope and provision of pharmacy based-harm minimisation services, (ii) complexity of service provision, and (iii) person-centred advice and support. Verbatim quotes are used to contextualise the data, with the following identifiers: stakeholder (S), pharmacist (P) or intern pharmacist (IP), pharmacy assistant (PA), and client (C).

### 3.1. Scope and Provision of Pharmacy-Based Harm Minimisation Services

Harm minimisation was broadly defined in terms of reducing patient harm from medication and drug and alcohol use, with stakeholders providing further detail on the three facets of Australia’s harm minimisation policy (i.e., demand, supply, and harm reduction). Pharmacy staff and stakeholders identified a range of available community pharmacy services, such as NSP and OTP. Pharmacists recognised staged supply as another service, with the introduction of pill testing raised by one pharmacist. More broadly, some pharmacists incorporated quality use of medicines services within the scope of pharmacy-based harm minimisation, including the return of unwanted medicines, dose administration aids, and in-store medication reviews, as well as checking for medication adherence and overuse. Consumers focused on OTP and NSP availability, with limited information received about safe injecting practices and disposal mechanisms. One pharmacist explained that they were concerned about endorsing illicit drug use if they provided information on safe injecting techniques, although agreed that this was an aspect of harm minimisation. Overall, a transactional approach for the supply of sharps kits was described:

“They just wanta, put it [sharps kit] in a bag and get you out the door really [laughs].”(C5, Male)

“…most of the customers they usually come in and they will have a low voice, can I please have a sharps kit and then they want to get it and then go…They don’t want to listen…it’s pretty hard for the front shop people or the pharmacist to even intervene with that.”(IP1, Male, Pharmacy 1)

Stories of being unable to obtain sharps kits from pharmacies were recalled, with two consumers suggesting that these be readily available or for staff to advise consumers of alternative access points. Pricing discrepancies across pharmacies were noted and one of the main reasons why consumers accessed other community-based NSPs, i.e., for free supply. Further promotion and information on pharmacy-based sharps disposal, not sharing needles and associated risks, and rehabilitation or other non-government support services was suggested by participants across the three groups. While some pharmacists confirmed that they would not know where to refer consumers to for further support, others believed that consumers were not interested in advice:

“…the kits don’t even have water in them and you’re trying to get them to buy water…We sell 100 kits a week but we might sell ten waters…so, I can’t even imagine then wanting to talk to them about, you know anything else, brochures and stuff.”(P4, Female, Pharmacy 3)

Two stakeholders raised concerns about limited naloxone stock availability within pharmacies. Consumers referred to their local pharmacy and identified that the pharmacist was unfamiliar with naloxone; that they did not have any in-stock; or, alternatively, that naloxone was difficult to order from the wholesaler. Both consumers and pharmacists did not consider naloxone when discussing pharmacy-related harm minimisation services; when prompted, one pharmacist admitted being unfamiliar with this medication. There was evidence that some pharmacists perceived naloxone as only useful for illicit drug users, i.e., not for prescription opioid users, or that there was limited consumer demand:

“…Never been asked so it’s probably just something that we wouldn’t necessarily ever need”(P9, Female, Pharmacy 6)

Stakeholders acknowledged that pharmacists were underutilised and well placed to assess client health and wellbeing; they were an alternative access point for consumers who could afford to pay privately or who sought anonymity. Further assistance with pain management, oral health, referring consumers to government and non-government support organisations or drug and alcohol services, and conducting a brief needs assessment for infections were discussed:

“…talk to them about hep [hepatitis] C treatment, talk to them about whether they’ve had a test, do they know their status?… If they haven’t been treated maybe try and direct them to a local GP [general practitioner] who is prepared to prescribe. Or even talk to their prescribing GP and ask them why they haven’t considered treating them.”(S5, Male)

Pharmacists described the personal benefits associated with their role in harm minimisation in terms of job satisfaction and client rapport. One pharmacist with strong Christian beliefs believed he was in a privileged position to assist OTP consumers, and that greater support was warranted for this at-risk population:

“…we say, ‘oh, here’s nicotine patches, here’s things you can do’. We never just say—‘oh, go stop smoking’…and that’s like a legal addiction. Like, and how much more addictive are these illegal addictions, so how much more so do they need like a way to manage tapering off and coming off it?”(P6, Male, Pharmacy 4)

A pharmacy owner stated that they had a social responsibility to give back to the community, but also viewed harm minimisation services as a practical way to mitigate security risks:

“…providing things like sharps kits prevents us from being a target for hold-ups and thefts and robberies…the theory is that people don’t shit in their own nest…”(P5, Male, Pharmacy 3)

### 3.2. The Complexity of Service Provision

In terms of dealing with difficult customers using harm minimisation services, the potential impact on other clientele and security concerns were discussed, and the difficulty in navigating what can be a challenging topic for pharmacists and consumers was apparent. Medication dependence or overuse was a problematic area to address with consumers. Service provision was at times challenging in the pharmacy; time constraints, being unable to find a specific locum doctor with appropriate skills to prescribe, or to provide direct referral to other services with already long waiting lists, were key issues.

Pharmacists described being constrained in their ability to help consumers when they were not privileged to background and health information, and in some cases were “flying blind” in being able to make a judgement call with respect to safe medication supply. Addressing concerns with potential addiction or medication overuse was fraught with complexity. One pharmacist, whilst acknowledging that this was not an ideal situation, confirmed that he did not attempt to approach consumers about medication concerns as they were unlikely to engage in conversation. There was apprehension with coming across in an accusatory manner alongside recognition that consumers may have a health condition that they are trying to manage:

“…I find it difficult. I don’t like confrontation. especially ’cause a lot of them are dependent on it, and they say ‘I need it for sleep…I’m not addicted’…if I am concerned, I try and contact a doctor if possible but again, they’re hard to contact sometimes as well.”(IP2, Female, Pharmacy 5)

While getting to know a consumer was deemed valuable for understanding a person’s situation, one pharmacist explained that there was a risk of knowing too much; trying to address overuse concerns resulted in one client moving on to another pharmacy. Alternatively, consumers could be disadvantaged by being honest, resulting in a punitive model of care:

“…people tell their doctor or tell their treatment provider that they’re using substances, and instead of you know, I don’t know—dealing with that in a better way, they get reduced off the program and so that increases harms by people, you know, need to use or return to drug use…”(S1, Female)

Limitations to accessing healthcare records, while not a common discussion point, were identified. While positives of a centralised record were recognised, there were concerns from one stakeholder about the implications of this on prescribing practices and how consumers would be managed if there was perceived medication overuse, e.g., from doctor shopping. Fear or concerns with disclosing medication or drug use was evidenced by the actions of consumers who described utilising different pharmacies to obtain their medication and health-related needs, such as sharps kits. This avoided judgement and assumptions about substance use:

“Because I feel like they’d look at me and think you’ve got mental health problems because you use drugs. Whereas it’s the other way round, I’ve got mental health problems so I use drugs. Probably doesn’t make it any better, but, that’s that…”(C3, Female)

“…there was a fear around that if he was scripted naloxone from the same pharmacy that he was getting his S8 [Schedule 8, controlled drug] medication from-that they would, I guess—cause a barrier for him in getting that medication.”(S3, Female)

There was acknowledgement from pharmacists that this situation was not ideal, that it was helpful to know the full picture to advise consumers of any potential drug-related interactions. Another pharmacist queried whether they were even allowed to disclose their concerns about other substance use to another health professional, for example, someone known to be on mental health medication requesting a sharps kit. Some pharmacists discussed that they had limited support or feedback from some prescribers in relation to their concerns, which made it difficult to make decisions regarding medication supply:

“…don’t seem to have that level of support from the private doctors, it’s pretty much ‘oh no I’ve written his script this month and I don’t want to see him for another month and I don’t care what he’s doing’…”(P2, Male, Pharmacy 1)

Managing situations when an OTP client could not pay for their dose was problematic. A stakeholder queried what would happen in this situation and suggested that direct-debit payment from Centrelink (an Australian government service that facilitates payment support for Australian residents, e.g., students, unemployed persons, retirees, and associated support services) could resolve this problem. Some participants perceived client charges as a barrier to treatment, with suggestions that this could be reduced:

“…we charge like $7 per day or $5 a day if they want to pay for a week…maybe like $2 per day would be more reasonable. Like just thinking about myself having to pay for that, that’s expensive.” (P3, Female, Pharmacy 2)

This participant felt conflicted knowing that OTP medication was supplied to pharmacies for free but consumers were charged dispensing fees. Another pharmacist stated that fees should be voided in legitimate situations, but that this placed value on the service. Alternative suggestions included subsidising OTP medication for Australian residents on the PBS, including buprenorphine once-monthly injections. Other comments made by pharmacists were in relation to time in providing these services, such as the associated paperwork and impact on workflow:

“…I’ve got one guy this morning and he’s on six medications. They’re all staged supplies and they’re all on different days and he requests an early supply all the time for all of them. And he’s a general patient and he doesn’t have the money to pay for his pickup, so the amount of time we spend on this particular patient-phenomenal…”(P4, Female, Pharmacy 3)

While one pharmacist indicated that they expected to spend some time in delivering these services which were remunerated, the lack of time was an impediment to providing person-centred care.

### 3.3. Person-Centred Advice and Support

Assumptions were made about the types of consumers who requested harm minimisation services. This included that only illicit drug users would require or need to use naloxone, and sharps kits were for the sole purpose of illicit drug use. Judgement towards this specific population was acknowledged; one pharmacist questioned his ability to trust OTP consumers and that this, alongside some security concerns, explained why this service was not offered. Stories about consumers feeling stigmatised or pharmacy staff interacting with these consumers differently when compared to other pharmacy consumers were shared:

“…you ask for a Fitpack *[sharps kit]* and they go ‘oh, one Fitpack’s $^1^0′ like really loud and there’s other people in there and, you know they look at you to see if you look wasted or not…”(C5, Male)

“…I just look at them and then go and get their dose. Do you know what I mean? Whereas everybody else who comes in, you’ll say ‘good morning.’”(P1, Female, Pharmacy 1)

Another pharmacist recalled that at university, his peers expressed a preference for undertaking placements at pharmacies that did not offer the OTP, a service considered unglamorous and difficult. Yet, undertaking a placement in an alcohol and drugs clinic was described by one pharmacist as an eye-opening and useful experience. This pharmacist gained some understanding of the patient’s lived experience, a side not often seen in the pharmacy setting:

“…they were like opening up about their lives and how sad they are and how one thing after another has led to them just being trapped in their addiction, and you always kind of know that that’s what happens, but you never really hear people really open up about it”(P3, Female, Pharmacy 2)

There was evidence of friendly, supportive care being provided by some pharmacists, such as the participant who lent one client movies so that he could catch up with films that he had missed while in jail, and the pharmacist who told a client that if they could not afford a sharps kit that they would supply it for free. This participant continued to describe that pharmacy consumers should be treated equally:

“I think some staff that we have would be on the conservative side to think that we shouldn’t be welcoming that type of patient into the pharmacy. My answer to that would be what type of patient do you refer to? We have all ranges of different patients that use this service and they shouldn’t be discriminated against just because they need the service”.(P8, Male, Pharmacy 5)

The impact and/or importance of privacy and confidentiality in providing harm minimisation services were acknowledged by all participant groups. The provision of OTP varied between pharmacies; some used a private area for dosing while others did not. There were mixed opinions as to whether a private area was needed; this was seen to either increase or reduce stigma-related concerns for consumers and/or other consumers:

“Some pharmacies’ dosing points are much more consumer-friendly than others…I know one or two pharmacies where the clients feel stigmatised by where they have to dose, the way they’re treated when they’re being dosed. You know, they’re kind of like put to the end of the line sort of thing and they’re in public and so it’s not ideal...”(S4, Male)

“…some *[OTP]* patients really like, like the privacy over there, but some patients feel a bit like they’re being treated like a druggo or like they’re a dangerous person… ”(P3, Female, Pharmacy 2)

Respecting privacy and confidentiality were key issues raised by one client, who explained that in the past, staff had disclosed details of their treatment to family members. This action was considered inappropriate from a health professional and a breach of a pharmacist’s duty of care. Stakeholders emphasised the importance of friendly, non-judgemental and accessible services that involved consumers in their own care and decision making that took a holistic approach to care:

“…You know, [staff] that aren’t going to tell you to stop using, you know. We can’t help you unless you make changes to your drug use or we can’t help you unless you do this.”(S1, Female)

“…if a pharmacy asked me about my sleep—because I’m basically an insomniac (laugh)—um, yeah I’d be willing to say ‘yes I do, I’ve got serious issues,’ but I wouldn’t go down there specifically to say ‘look I’ve got this sleeping issue’…if somebody came up to me and said ‘you look really, really tired, is there something that we can help you out with?’ yeah I think I’d be more willing to open up.”(C4, Male)

## 4. Discussion

This study involved interviews to explore opinions about the role of community pharmacy in provision of harm minimisation services. Interviews with 28 participants showed scope for expansion of the role of community pharmacy whilst also highlighting the complexities involved in service provision, with room for improvement in some areas.

Community pharmacies could be viewed as well placed to support harm minimisation services as they are widely distributed and accessible healthcare destinations. Some interview stakeholders believed that community pharmacies were underutilised in the provision of harm minimisation and related patient care services, such as support with pain and oral health management and monitoring for infections. Despite these opinions from stakeholders, interviewed consumers focused solely on the role involving OTP and NSP. This narrow view of the role of community pharmacy is similar to the findings of the 2012 harm minimisation literature review that identified roles in OTPs and provision of sterile needles [13]. However, new community pharmacy services and roles had since become available and been implemented [55].

Some pharmacists described personal benefits from providing harm minimisation services in terms of increased job satisfaction and client rapport, which are similar findings to an Australian study that explored community pharmacists’ role in provision of enhanced and extended services [56]. Being supportive and non-judgemental were viewed as essential by pharmacists, and stakeholders emphasised the importance of accessible, friendly, and holistic services that incorporate patient-centred care. However, stigma was identified as a barrier, with pharmacy staff and consumers sharing stories about consumers feeling stigmatised or staff interacting with these consumers differently compared to other pharmacy consumers. These barriers are not unique compared to other study findings on the implementation of harm-minimisation services in community pharmacies, both in Australia and internationally [29,30,32,40,41], and must be addressed for the profession to take on a bigger role in harm minimisation.

Interviews revealed that pharmacy staff had little collaboration with organisation(s) or services that assist consumers with drug misuse, with only few having a working relationship with these organisations or services. A small number reported that they were aware of referral services in case of an overdose, indicating a need for staff to be educated about support services. Consumers specifically identified lack of information from pharmacy staff about support organisations, safe injecting practices, and mechanisms for the safe disposal of sharps. The approach reported by some consumers and pharmacy staff was that of a transactional nature during the supply of sharps kits, which could be related to stigmatising attitudes or a lack of knowledge or confidence. A recent U.S. study that focused on the supply and advice provided by pharmacists on the use of naloxone showed pharmacists’ lack of knowledge of harm minimisation services were the main barriers to the implementation and promotion of these services [40]. Self-confident, well-trained pharmacists should be able to proactively offer public health services that are likely to improve consumer attitudes and health [57].

The importance of privacy and confidentiality was acknowledged by all interview participant groups, with one consumer raising an incident with staff disclosing treatment details to family members. Addressing privacy and confidentiality concerns is crucial, with UK and Australian research highlighting that lack of privacy and confidentiality inhibits service utilisation [58,59]. An Australian study found that the use of private consultation areas for certain services and sensitive discussions was supported by pharmacists and consumers, although there was recognition for the need to incorporate this into workflow processes [60]. Strategies used by staff to overcome privacy obstacles included taking consumers to a quieter part of the pharmacy, lowering of voices, and interacting during pharmacy quiet times [60]. All of these strategies should be used by pharmacy staff in the provision of harm minimisation services. Although some interview pharmacy staff used a private area for dosing of OTP others did not. However, there were mixed opinions as to whether a private area was needed as this was seen to either increase or reduce stigma-related concerns for OTP consumers and other consumers. Of interest was the consumer comment that this retail environment provided a degree of anonymity, further highlighting pharmacy as an alternative option for consumers to seek healthcare advice, even with the associated privacy challenges.

Some pharmacists commented on the constraints involved in managing consumers’ medicines if not privileged to background and health information. Lack of information on diagnosis and management impacts on pharmacists’ ability to provide patient-centred care. Some consumers, on the other hand, had fears and concerns with disclosing medication or drug use information with pharmacists, evidenced by them describing using different pharmacies instead of obtaining their medication and health related needs, such as sharps kits, from one location. This avoided judgement and assumptions by pharmacists about substance use. Research that focused on the role of community pharmacists in mental health showed there were trusting relationships between pharmacy staff and consumers, which were identified as important in creating safe healthcare spaces [61,62]. There is scope to better inform the users of community pharmacy harm minimisation services about pharmacists’ confidentiality and information sharing professional obligations, as underpinned in the various codes of ethics and conduct [63,64].

Financial issues were raised by consumers as well as pharmacy staff. Pricing discrepancies across pharmacies was one of the main reasons why consumers accessed other community-based NSPs for free supply. From the pharmacy staff perspective, tensions were identified with existing commercial business demands and the low remuneration received for harm minimisation services. Other financial concerns raised during the interviews were managing situations when OTP consumers were unable to pay for their methadone or buprenorphine doses. Other studies have identified similar challenges with OTP payments, showing that daily methadone dispensing fees adversely impacted on patient finances resulting in treatment cessation [24] or skipping doses as a consequence of being unable to pay on time [25]. There have been ongoing calls for the Australian federal government to fully subsidise the OTP, not only from the pharmacy profession but also GPs and addiction experts, quoting that the costs are preventing more people from accessing treatment [65]. Addressing the various cost issues from both the pharmacy staff perspectives in terms of workforce and from the consumers’ perspectives for out-of-pocket costs are crucial in facilitating community pharmacy harm minimisation services.

### Limitations

The limitation of the study was the small number of interview participants who were identified using snowball sampling from prior researcher knowledge. This has the potential to lead to a selection bias of participants with an interest and knowledge in harm minimisation, and thus may not reflect a balanced view in this sector. Furthermore, the study was conducted in QLD; the findings cannot be generalised to represent the overall landscape on the delivery of harm minimisation services in community pharmacies across Australia.

## 5. Conclusions

Community pharmacies remain a valuable healthcare destination that could expand delivery of harm minimisation services, albeit with some improvements from within the profession and health sector. There is a need for better strategies at local and national levels to facilitate the harm minimisation role of community pharmacies. Community pharmacists have an opportunity to engage consumers requiring support in harm minimisation, explaining their professional obligations to confidentiality and to foster a trusting relationship. Better collaboration and sustained linkage between pharmacies and relevant organisations or providers in this sector is vital for the provision of holistic care for these consumers.

## Figures and Tables

**Table 1 pharmacy-09-00095-t001:** Interview guide.

Main Focus	Interviewer Questions
Background	Briefly outline your current position and what this involves. Does your organisation provide any direct services/support for consumers with addiction/s? Please explain.
Harm minimisation: concept and related services	What does the term harm minimisation mean to you? What is your/organisation’s perspective on harm minimisation services? What services do you/organisation believe are part of an approach to harm minimisation in the community? Why?
Role of community pharmacy	What is your/organisation’s perspective on the services located within the community pharmacy? How could community pharmacy help consumers with addiction (as part of a public health role)? Why? What pharmacy services are related to harm minimisation? Describe the services your community pharmacy provides for patients/caregivers with an addiction. Do you promote harm minimisation services in the pharmacy setting? What are the benefits of providing harm minimisation services in community pharmacies? What are the risks? Barriers? What can community pharmacy staff do to better support you [consumer] in relation to your lifestyle choices? Do you [consumer] go to the community pharmacy for other health-related services/has this ever been offered to you? What influences your [consumer] choice of pharmacy? What do you perceive to be your role [pharmacist/assistant] in relation to harm minimisation?
Pharmacy experiences	What do your members think about community pharmacy and the services they provide?Why? Have any customers/clients enquired about a service that could be potentially offered within a pharmacy? Describe your experiences accessing a harm minimisation service within community pharmacy (including privacy). What information have you ever received from a pharmacy in relation to harm minimisation? Can you tell me about your experiences with consumers with addictions in the pharmacy? If you could change anything about the interaction/s you have had with pharmacy staff, what would this be and why? Describe what your ideal community pharmacy service would look like for consumers with an addiction/s? Can you tell me about your experiences with consumers with addictions in the pharmacy? Have you identified any differences in the services provided for consumers/caregivers with an addiction/s as compared to physical illness? To other mental illnesses?
Managing addiction	Do you use a needle and syringe service? Where do you access this service? Why? How would you describe your confidence in your ability to speak with your consumers about addiction concerns? What is the typical plan of action when you encounter a patient you perceive to be addicted to medicines? Think of a time when you should have engaged a patient in a conversation about addiction but chose not to do so/did so. Please describe the situation. Describe the healthcare team members you work with in providing your services to consumers/carers with an addiction/s.

## Data Availability

The data presented in this study are available upon reasonable request from the corresponding author. The data are not publicly available due to ethical reasons.

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
