# Peer review of "Australian Community Pharmacy Harm-Minimisation Services: Scope for Service Expansion to Improve Healthcare Access"

_pharmacy, 2021, doi:10.3390/pharmacy9020095_

Round 1

Reviewer 1 Report

This is a well written article. Please kindly improve the introduction, for those countries who may not understand how, Australia, USA, UK pharmacy works.

I would kindly ask to present this specific service in a chart form, step by step intervention. Please mention about the traning and certification for this service.  I think it would give a better look to pharmacists in some countries who do not know about metioned intervention in the article.

You are right about the sample size in the limitation section, it should be improved in the future. I suggest, to call it a pilot study. Overall, the presented work is very interesting and neatly prepared.

Author Response

Response to Reviewer Comments Pharmacy-1163980

Title: Australian community pharmacy harm minimization services: scope for service expansion to improve healthcare access

The authors wish to thank the Editor and the Reviewers for taking the time to review the submission. The authors have carefully considered the comments of the Reviewers and suggestions and made necessary changes to improve the manuscript. Detailed responses are below with blue text indicating amendments to the manuscript.

Comment

Response

Reviewer 1

This is a well written article. Please kindly improve the introduction, for those countries who may not understand how, Australia, USA, UK pharmacy works.

Thank you for the positive feedback.

Details on the dates the various harm minimization services were implemented were added to the Introduction to provide better context to the various services being available in Australia. Line 50: The NSP has been available in Australia since the 1980s and involves provision of sterile needles and syringes and safe disposal methods to reduce the transmission of blood-borne viruses and other infections caused by non-sterile practices[12-14].

Line 55: The OTP has been in place in Australia since the 1970s and aims to reduce opioid misuse through the supply of subsidized opioid medicines, i.e. methadone and buprenorphine, to reduce cravings, relapse risk and withdrawal symptoms

Line 78-80: Naloxone was rescheduled in Australia in 2015 to become available as a pharmacy OTC medicine that does not require a prescription.

Line 90-91: Staged supply is another harm minimization service provided by Australian community pharmacies that was introduced in 2011 and funded by the Federal Government

We considered including further information on how the health and pharmacy sectors work in the various countries but felt it was outside the scope of this paper.

I would kindly ask to present this specific service in a chart form, step by step intervention. Please mention about the traning and certification for this service.  I think it would give a better look to pharmacists in some countries who do not know about metioned intervention in the article.

Thank you for this comment. We decided not to go into extended details about certification for the various harm minimization services as it varies between Australian jurisdictions and there is no actual training or certification needed for pharmacists to provide harm minimization services. It is part of the scope of practice of community pharmacists to provide these services.

You are right about the sample size in the limitation section, it should be improved in the future. I suggest, to call it a pilot study. Overall, the presented work is very interesting and neatly prepared.

This study was a qualitative exploratory study to inform the development of a survey and hence not a pilot study.

Reviewer 2 Report

The scope of the current work - understanding attitudes towards harm minimization generally including specific interventions (needles, OTP) as well as cost/funding issues is very broad, and a challenge throughout the manuscript including the introduction and results. The work is very practical and seems very important, to Australian pharmacy but also internationally.

For an international reader, a brief history (including some dates/years) would be useful to set the stage for understanding where Australia is in terms of harm minimization implementation. E.g. “needle provision from pharmacies was expanded in 20XX”, “naloxone became available from community pharmacies in 20XX” etc. This would help the reader place some of the results and interview statements in context. Similarly, opioid overdose #s are indicated but the reader could use information on trends, whether they are going up, etc.

Stigma is addressed in both the introduction, results, and discussion but appears in several places in each. However, stigma seems to underly many of the issues and comments throughout the results section. I recommend the authors consider whether the issue of stigma should be addressed more systematically, e.g. its own paragraph in the intro, its own distinct paragraph(s) in the discussion.

Similarly, financial issues related to patient cost, government cost are also found in a couple places throughout the introduction and scattered throughout the results. It might be useful to include a stand-alone paragraph about cost in the intro, or referring the reader to a government resource that summarizes current cost/renumeration for harm minimization-related services

The literature review and results reflect what seems to be evolving attitudes and approaches to harm minimization. Uncertainty is conveyed by stakeholders but in particular pharmacists and consumers with respect to many of the issues. This and other work could support a call for more leadership in this area, perhaps national or regional “harm minimization strategies” if those don’t already exist. The conclusion section could be used to make a more direct and  provocative call for such strategies.

Provide more information on key stakeholders - are they decision makers? advocates? academics?

Are pharmacy owners necessarily pharmacists?

Author Response

Response to Reviewer Comments Pharmacy-1163980

Title: Australian community pharmacy harm minimization services: scope for service expansion to improve healthcare access

The authors wish to thank the Editor and the Reviewers for taking the time to review the submission. The authors have carefully considered the comments of the Reviewers and suggestions and made necessary changes to improve the manuscript. Detailed responses are below with blue text indicating amendments to the manuscript.

Comment

Response

Reviewer 2

The scope of the current work - understanding attitudes towards harm minimization generally including specific interventions (needles, OTP) as well as cost/funding issues is very broad, and a challenge throughout the manuscript including the introduction and results. The work is very practical and seems very important, to Australian pharmacy but also internationally.

For an international reader, a brief history (including some dates/years) would be useful to set the stage for understanding where Australia is in terms of harm minimization implementation. E.g. “needle provision from pharmacies was expanded in 20XX”, “naloxone became available from community pharmacies in 20XX” etc. This would help the reader place some of the results and interview statements in context. Similarly, opioid overdose #s are indicated but the reader could use information on trends, whether they are going up, etc.

Thank you for the positive feedback and for pointing out how to contextualise the manuscript better for international readers. We have added the following text to indicate when the various services were introduced with references:

Line 50: The NSP has been available in Australia since the 1980s and involves provision of sterile needles and syringes and safe disposal methods to reduce the transmission of blood-borne viruses and other infections caused by non-sterile practices[12-14].

Line 55: The OTP has been in place in Australia since the 1970s and aims to reduce opioid misuse through the supply of subsidized opioid medicines, i.e. methadone and buprenorphine, to reduce cravings, relapse risk and withdrawal symptoms

Line 78-80: Naloxone was rescheduled in Australia in 2015 to become available as a pharmacy OTC medicine that does not require a prescription.

Line 90-91: Staged supply is another harm minimization service provided by Australian community pharmacies that was introduced in 2011 and funded by the Federal Government

The following statement was added with two references, line 37:

Trends of opioid overdoses are variable and fluctuate [6,7].

6.            Scholl L; Seth P; Kariisa M; Wilson N; G., B. Drug and Opioid-Involved Overdose Deaths — United States, 2013–2017. MMWR Morb Mortal Wkly Rep. 2019, 67, 1419–1427.

7.            Centers for Disease Control and Prevention; National Center for Injury Prevention and Control. Prescription Opioid Overdose Death Maps. Available online: https://www.cdc.gov/drugoverdose/data/prescribing/overdose-death-maps.html (accessed on 23.04.21).

Stigma is addressed in both the introduction, results, and discussion but appears in several places in each. However, stigma seems to underly many of the issues and comments throughout the results section. I recommend the authors consider whether the issue of stigma should be addressed more systematically, e.g. its own paragraph in the intro, its own distinct paragraph(s) in the discussion.

Thank you for the comments. We have assessed combining of the statements about stigma, currently in lines 74 and 90 in the introduction and 417-423 and 430-432 in the Discussion. Considering that the manuscript flows well and the word count, we decided to leave it as is.

Similarly, financial issues related to patient cost, government cost are also found in a couple places throughout the introduction and scattered throughout the results. It might be useful to include a stand-alone paragraph about cost in the intro, or referring the reader to a government resource that summarizes current cost/renumeration for harm minimization-related services

Thank you for the comments. As with the comment above, we have considered the statements, currently lines 64-70, 84-86, 89 and 92 of the Introduction and the last paragraph of the Discussion (lines 468-482).

There is no single resource that outlines the cost for harm minimization services and pharmacies can charge what they deem to be appropriate. The 6th Community Pharmacy Agreement provides some guidance on how to claim money back from the Commonwealth Government for Staged Supply services at

https://6cpa.com.au/6cpa-programs/medication-adherence-and-medication-management-programs/

We have not included this reference as it only covers one harm minimization service and it refers to the process for pharmacies to claim money and not the cost involved from consumers. We have therefore decided to not include this in the manuscript.

The literature review and results reflect what seems to be evolving attitudes and approaches to harm minimization. Uncertainty is conveyed by stakeholders but in particular pharmacists and consumers with respect to many of the issues. This and other work could support a call for more leadership in this area, perhaps national or regional “harm minimization strategies” if those don’t already exist. The conclusion section could be used to make a more direct and  provocative call for such strategies.

The following sentence was added to the Conclusion to emphasise the need for local and national strategies:

There is a need for better strategies at local and national levels to facilitate the harm minimization role of community pharmacies.

Provide more information on key stakeholders - are they decision makers? advocates? academics?

Thank you for highlighting that this needed to be added. The following sentence was added lines 180-181:

Stakeholders represented staff and clinicians in policy positions from both government and non-government organisations.

Are pharmacy owners necessarily pharmacists?

Yes, pharmacy owners in Australia need to be pharmacists as per the first paragraph of the Results:

Most pharmacy staff interviewed were registered pharmacists (n=11), of which six were female and three were pharmacy owners.